# It's not you, it's me: A thematic analysis of written causal attributions for favorable and unfavorable feedback on different types of Instagram image sharing

**Malinda Desjarlais**[ID]*

Department of Psychology, Mount Royal University, Calgary, Alberta, Canada

* mddesjarlais@mtroyal.ca

## Abstract

Previous research has provided valuable insights into how virtual likes and comments on self-presentational content affect users' emotions and self-evaluations, yet less is known about the causal attributions people make in response to such feedback. This study analyzed responses to an open-ended question to explore the types of causal attributions individuals make for favorable and unfavorable feedback on different types of Instagram posts – selfies, elsies (self-images taken by others), and memes. Undergraduate participants (N = 125; 77% female; 81% 17- to 24-years-old) were randomly assigned to one of three image conditions (selfie, elsie, or meme) and told their selected image would be posted to a university-affiliated Instagram account. After a 5-minute distraction period, during which participants believed their image was posted for feedback, they were told they had received either favorable or unfavorable (fabricated) feedback. They were asked to write down the perceived causes of the feedback they received in an open-ended question. Additionally, participants reported their perception of societal attitudes toward selfies compared to elsies, to explore whether these perceptions may have influenced their attributions. Findings revealed that self-focused images (selfies and elsies) more frequently prompted internal attributions – especially appearance-related – compared to meme posts, which elicited primarily external attributions tied to audience reception. Despite recognizing society's negative perceptions of selfie-posters, participants rarely attributed feedback to these broader social norms. The results suggest that posting self-images can heighten self-awareness and self-objectification, leading individuals to internalize social feedback. These attribution patterns carry implications for self-esteem and self-presentation. Understanding how users interpret feedback may inform efforts to promote healthier engagement with social media.

**Data availability statement:** A copy of the anonymized data for transparency purposes has been made available at: https://doi.org/10.17605/OSF.IO/JR379.

**Funding:** The author(s) received no specific funding for this work.

**Competing interests:** The author has declared that no competing interests exist.

## Introduction

Individuals engage in selective self-presentation by highlighting desirable aspects of themselves and downplaying undesirable ones [1]. In turn, receiving compliments, praise, and expressions of liking can boost self-esteem, whereas criticism and rejection may lower it [1]. Becoming ubiquitous in daily life [2], social networking sites (SNSs) such as Instagram, provide a unique context in which self-presentation and peer influence interact to co-construct identity. Instagram offers opportunities for individuals to share information about themselves through visual and textual content, and to receive feedback in the form of virtual likes and comments. Unlike offline interactions, online self-presentations are typically shared with broad audiences and become persistent artifacts. This content can be viewed repeatedly by diverse individuals over time, with public and enduring responses that may further shape how the self is presented on SNSs [3]. Individuals tailor their self-presentation to align with perceived expectations and preferences [4,5], not only for those who directly engage with the content but also for imagined audiences and their internal audience (i.e., their own knowledge and standards shaped by past experiences) [6]. This gives rise to the question, how do individuals negotiate self-presentation and social perception on Instagram?

### Theoretical framework

Individuals are rather selective in the photos they share on Instagram, as such images convey various aspects of who they are, including their appearance, personal style, experiences, and humor, and are broadcast to merged social networks [3]. Selfies and elsies are a prevalent form of self-presentation on Instagram. A selfie is a photograph that individuals take of themselves, usually holding the camera at arm's length or pointed at a mirror. In a survey of 18- to 24-year-olds, 98% reported taking selfies, 46% shared one within the past day, and 69% shared selfies three to 20 times per day [7]. Like selfies, elsies are self-images; however, they are captured by another person rather than the individual depicted. These images tend to adopt a candid style that generally portrays everyday life from the observer's, rather than the actor's, perspective [8]. Although both selfies and elsies feature the individual as the focal point, they are often perceived quite differently. Selfies are regarded as less authentic and engaging than elsies, and selfie-posters are judged as more narcissistic and less reserved than those who share elsies [9–13].

In addition to curating their self-presentation on SNSs, individuals receive feedback in the form of virtual likes and comments, which elicit emotional and psychological responses that in turn influence subsequent self-presentation. Although virtual likes represent a lightweight form of communication, many users attribute significant meaning to this seemingly simple gesture. For some, the number of likes received functions as a proxy for peer attention, a metric for social comparison, and a reflection of their social appeal, often carrying more weight than the nature of the comments themselves [4,14]. Both the quantity of likes [14,15] and the tone of comments [16,17] have been linked to fluctuations in self-esteem. A lower-than-expected

number of likes often evokes strong negative emotional responses [5], especially among younger females and individuals with low self-esteem [4,18,19, 20]. As a result, self-presentation is often shaped by users' expectations about what content will elicit a satisfactory response from their social network [4,5].

This pattern has been interpreted by researchers through the lens of sociometer theory [14,18,21,22], which conceptualizes self-esteem as a psychological gauge of social acceptance. Within this framework, self-esteem fluctuates in response to perceived cues of inclusion or rejection. Since virtual likes are treated as symbolic indicators of social approval, researchers argue that receiving fewer likes than expected is interpreted as a sign of social rejection, resulting in a decline in self-esteem [18]. However, while much research has focused on the emotional and self-esteem-related consequences of such feedback, relatively little is known about the causal inferences individuals make in response to it. Specifically, what do individuals infer as the reason behind receiving social approval or disapproval? The attributions people assign to favorable and unfavorable social media feedback—those underlying or accompanying emotional reactions—remain underexplored.

Weiner's causal attribution theory provides a useful framework for understanding how individuals interpret the social feedback they receive in response to what they share on Instagram. Accordingly, people try to make sense of others' behavior by identifying why something happened and who or what is responsible [23,24]. Causal attributions are classified along three key dimensions: locus (whether the cause is internal or external), stability (whether the cause is consistent over time), and controllability (whether the outcome is under the individual's control). Internal factors refer to characteristics within the person, such as traits and intentions (e.g., *I* am not very attractive), whereas external factors include the context, audience characteristics, and prevailing social norms (e.g., *People* are generally nice). Together, these dimensions shape how people interpret social feedback and, in turn, their self-perceptions and emotional responses [23,24].

Gaining insight into the attributional processes individuals infer for favorable and unfavorable feedback may be critical for understanding more nuanced links between SNS use and mental health. Although attribution theory was not their guiding framework, one study provides useful insights into the reasons individuals consider when interpreting audience responses to online self-presentation. In their interview study, Stsiampkouskaya and colleagues asked participants to imagine a couple going on camping trip, select a photo the couple might post to social media, and then reflect on how the couple may respond to alternative feedback-related scenarios [5]. Aligning with the extant literature, emotional reactions followed a clear pattern: happiness emerged from perceptions of successful impression management, while disappointment was tied to perceived failure. When feedback fell short of expectations, participants expressed disappointment and often attributed the outcome to a failure to capture attention or generate interest – treating the photo like a piece of art that failed to intrigue, implying the issue lay in the content, not the individuals themselves. Other explanations pointed to contextual or situational factors, such as the post being unintentionally set to private, the audience being preoccupied by offline activities, or competing content posted simultaneously. Notably, participants rarely cited flaws in the photo itself as the reason for low engagement, reinforcing the tendency toward external attributions. Conversely, when imagining scenarios where the photo received more feedback than expected, participants viewed it as successful impression management – their audience was impressed, and the content was engaging. At the same time, they often downplayed the significance of this success by attributing it to external circumstances, such as the audience being in a particularly receptive mood, rather than stable, internal qualities of the individuals. This suggests that even favorable outcomes were sometimes viewed as situational rather than self-reflective.

Self-awareness is a key mechanism that shifts the spotlight of attention from external events to internal states [25]. Self-awareness refers to the psychological state in which individuals focus attention on themselves, becoming aware of their thoughts, feelings, behaviours, and traits. Particularly, a focus on how one appears to others is triggered when one feels observed – as would be the case when posting self-images on SNSs for feedback [26]. In their study, Stsiampkouskaya and colleagues had participants adopt a hypothetical identity and select posts on behalf of fictional individuals [5]. Because the images did not feature the participants themselves, they were able to psychologically distance their own

internal attributes from the imagined feedback. As a result, it remains unclear how well these attributions reflect individuals' real-life inferences when social media successes and failures are personally relevant.

The present study extends this work by comparing selfies and elsies, which tend to heighten self-awareness and direct attention inward, to memes, which divert focus outward. Memes are a widely used form of humorous visual content, with three-quarters of 13- to 36-year-olds reporting they have shared a meme and just over half indicating they share them weekly [27]. Although typically shared to entertain, memes can also be used to express emotions, beliefs, and personality traits, serving a form of self-presentation characteristics. They also offer a form of self-presentation that does not emphasize physical appearance. Examining how individuals interpret other's favorable and unfavorable responses to selfies, elsies, and memes through the lens of the causal attribution theory can offer valuable insight into how identity, self-presentation, and social perception are negotiated in digital contexts.

### Current study

Employing an experimental design in which participants were led to believe that their selfie, elsie, or selected meme had received either favorable or unfavorable feedback, the present study seeks to advance our understanding of social feedback processes by examining: *What attributional explanations do individuals formulate for the social responses their posts elicit, particularly with respect to the locus-of-control dimension?* Furthermore, to contextualize potential attributional patterns and extend existing research on personal evaluations of others' self-presentation, the study also investigates a secondary question: *What are individuals' perspectives of society's opinions toward those who post selfies versus elsies?*

### Method

Mount Royal University Human Research Ethics Board has approved the design of this research (Application #101329). Written consent was obtained both before completion of the study protocol and after debriefing when they were informed of the deception involved and the true purpose of the study.

### Participants

Initially, 127 participants completed the study, however scores from two individuals had to be removed because of their accurate suspicion of deception. The final sample consisted of 125 undergraduate students (97 females, 27 males, and 1 non-binary) with a mean age of 21.4 years ($SD = 4.4$, range 17–43 years, 81% 17- to 24-years-old) who volunteered to participate for course credit. All participants reported using SNSs, with daily estimated usage ranging from just under 5 minutes to over 8 hours ($M = 2.7$ hours per day, $SD = 2.0$). Instagram was the platform most commonly visited on a regular basis, with 91% of participants reporting regular use. On a scale from 1 (never) to 7 (always), participants indicated that they posted selfies when visiting SNSs relatively infrequently ($M = 2.9$, $SD = 1.3$).

### Design

The study design was approved by the Human Research Ethics Board of the author's institution. Recruitment took place March 8, 2018, until March 13, 2020. Participants were randomly assigned to one of six conditions in a 3 (image type: selfie, elsie, or meme) x 2 (feedback condition: favorable or unfavorable) between-subjects design. Participants were asked to take a selfie ($n = 43$), have an elsie taken by the researcher ($n = 40$), or select a meme from a pre-determined set ($n = 42$). As not to bias the outcome, participants were informed that one purpose of the study was to examine reactions to different types of social media posts. They were told that the image they selected would be posted on the university psychology Instagram page for 5 minutes, accompanied by the hashtags #studentlife #hangingout, and the university acronym, allowing their peers to view and respond. Participants in the selfie condition were instructed to use the provided iPhone to take as many selfies as they wished, from which they would select one for the researcher to post. In the elsie

condition, participants were told that the researcher would take as many images of them as they desired, from which they would then choose one to be posted. Those in the meme condition were shown 10 different memes and asked to select one they most affiliated with – whether it captured their character, matched their sense of humor, or reflected their opinion – to be posted.

Participants were also randomly assigned to one of two feedback conditions: unfavorable (*n* = 63) or favorable (*n* = 62) feedback in response to their posts. In the unfavorable feedback condition, participants were informed: "Compared to results from our pilot study, your image has received far below the average number of likes. And there were some fairly negative comments." In contrast, those in the favorable feedback condition were told: "Compared to results from our pilot study, your image has received an above average number of likes. And there were some fairly positive comments." Providing feedback in the form of likes relative to the mean from a pilot study, rather than assigning a fixed number, was modeled after Burrow and Rainone [18]. This relative framing introduced ambiguity, encouraging participants to interpret the feedback as either favorable or unfavorable in comparison to others. Furthermore, referencing comments as positive or negative was informed by Valkenburg et al. [28], who found that the valence of comments has greater impact than their frequency.

## Measures

**Perceptions of societal attitudes.** Perceptions of societal attitudes were collected through two written open-response questions: (1) "Think about society's typical impression or view about people who share selfies on their social media pages. What is society's typically impression of people who post selfies?" and (2) "Think about society's typical impression or view about people who share on their social media pages images of themselves that others took (e.g., Suzy posts a picture of herself at a party that her friend Jen took). What is society's typical impression of people who post images of themselves that others took?" Perceptions were also assessed quantitatively asking participants to rate their perceptions of society's impression of those who post selfies and elsies on 7-point scales, anchored by: 1 = *Authentic/genuine* and 7 = *inauthentic fake*; 1 = *posters are attention seekers* and 7 = *posters are reserved*; and 1 = *attitudes about the image are very positive* to 7 = *attitudes are very negative*.

**Attributions of feedback.** As a manipulation check, valence of feedback was rated on a 7-point scale anchored by extremely negative to extremely positive in response to the question: "Which best describes the feedback you received from others about your image posted on social media?" To explore participants' attributions for their feedback, they were presented with the open-ended prompt: "What was one major cause of the likes and comments that you received?" Additionally, a quantitative measure was included to access attributions along the dimension of locus of control: "Is the cause of the likes and comments you received due to something about you or something about other people or circumstances?" Reponses were recorded on a 7-point scale ranging from 1 (*totally due to other people or circumstances*) to 7 (*totally due to me*).

## Procedure

Participants arrived at the lab and were guided through an informed consent process and subsequently provided written consent. In accordance with the institution's ethical guidelines, university students, irrespective of age, may independently provide informed consent for research participation. Accordingly, parental consent was not required for participants under 18 years of age. See Fig 1 for a summary of the procedure. Participants were presented with a cover story stating that the purpose of the study was to examine the implications of social media use for both those who post content and those who view it, and that their role would be that of the content poster. Participants then received their randomly assigned image condition instructions, wherein the photo-taking process (where applicable) and selection was completed. During the 5-minute posting span, participants completed a brief personality questionnaire (scores were not recorded) as a distractor task. Following this they were provided with either favorable or unfavorable feedback depending on the feedback

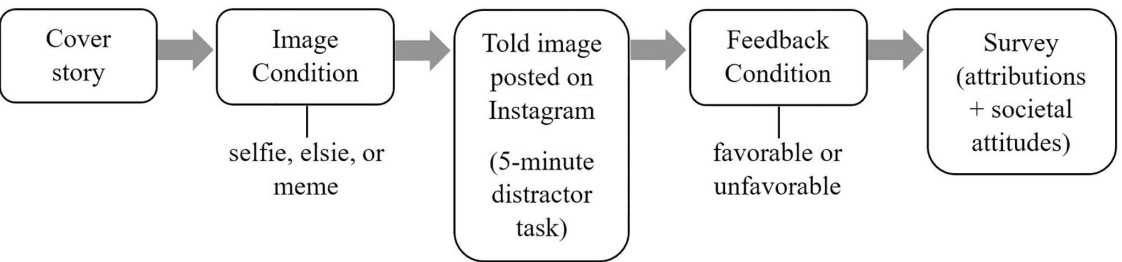

**Fig 1. Summary of study procedure.**

condition. Notably, images were never posted online, and feedback was fabricated. Participants then completed an online survey consisting of the items assessing causal attributions for the feedback received, followed by the questions about perceived societal attitudes toward selfies and elsies. Upon completion of the survey, participants were fully debriefed. They were informed of the true purpose of the study and the deception that ensued and asked to provide consent for their data to remain in the study knowing its true purpose.

## Analyses

Quantitative analyses were conducted using multiple *t* tests for dependent groups and a 3 (image condition) x 2 (feedback condition) between-subjects analysis of variance (ANOVA) to examine ratings. An alpha level of .05 was used for each analysis. One participant who did not provide a rating for the locus of control question was excluded from the corresponding analysis. These analyses were conducted using R version 4.5.0.

Qualitative responses were analyzed using reflexive thematic analysis, following the guidelines of Braun and Clarke [29]. I began by reading through all responses to gain familiarity with the data, intentionally bracketing out information about experimental conditions to reduce bias. Initial coding involved identifying discrete content categories within each response, with multiple codes applied when necessary to capture nuances. These codes were then reviewed and grouped into more abstract subthemes, which were subsequently refined into broader, overarching themes. This iterative process involved movement between the raw data and developing interpretations to ensure that themes remained grounded in participants' responses. All participants from the final sample were retained in the qualitative analysis, with sample sizes reported to indicate how may provided written responses to each open-ended question.

## Results

### Perceptions of societal views toward selfies and elsies

Themes for perceptions of society's view toward selfies (*n* = 119) and elsies (*n* = 116) are presented in Table 1.

On average, participants rated society's perceptions of individuals who post selfies (*M* = 4.7, *SD* = 1.3) as significantly more inauthentic/fake compared to those who post elsies (*M* = 2.8, *SD* = 1.3), *t*(124) = 13.29, *p* < .001. Notably, the mean rating for selfie-posters fell near the midpoint of the scale, suggesting a relatively neutral perceived societal attitude regarding authenticity. Although authenticity was not a prevalent theme, aligning with the quantitative findings, twice as many participants described society as perceiving selfie-posters as inauthentic rather than authentic. Their perceptions of societal attitudes toward elsie-posters were skewed in a more positive direction; nine participants described elsie-posters as being viewed as authentic or genuine, whereas only three noted they may be perceived as misleading or inauthentic.

On average, participants rated society's views of selfie-posters (*M* = 2.9, *SD* = 1.3) as significantly more likely to be attention-seeking compared to elsie-posters (*M* = 4.0, *SD* = 1.3), *t*(124) = −7.69, *p* < .001. Participants showed a clear consensus in the types of motives they believed society attributes to selfies, with 14 explicitly referencing attention-seeking

**Table 1. Themes for societal perceptions of selfie- and elsie-posters.** Note. Coding in parentheses identifies if the content was cited in reference to selfies (S) or elsies (E). Values in parentheses indicate the number of participants who referenced the content. Comments were provided by 119 participants for selfie-posters and by 116 participants for elsie-posters.

| Overarching Theme Subtheme | Content Category |
|---|---|
| **1. Presumed Motives** | - attention-seeking, validation (S14; E9)<br>- social comparison, self-evaluation (S3; E2)<br>- sharing about self (S1; E6)<br>- bragging (S3; E6)<br>- documentation (E1)<br>- curate self-presentation (E4)<br>- demonstrate beauty (E1) |
| **2. Broad appraisal** | - positive impression (E1)<br>- negative impression (E1) |
| **3. Character assessment** | |
| (a) Surface-level traits | - superficial (S1; E1)<br>- appearance-focused (S2; E1) |
| (b) Narcissistic traits | - general (S7; E1)<br>- self-centered, self-absorbed (E31; S1)<br>- conceited or vain (S13; E2)<br>- likes attention (E1)<br>- exhibiting superiority (E1)<br>- not self-absorbed (E1) |
| (c) Confidence | - confident, high self-esteem (E31; S12)<br>- insecure (S3)<br>- not confident (S1)<br>- too confident (E1) |
| (d) Affect | - happy (S1; E2)<br>- carefree (E1) |
| (e) Socialization | - social (S1)<br>- friendly, social, outgoing (E10)<br>- popular, lots of friends, well-liked (E16)<br>- reserved (E1)<br>- having fun (E5) |
| (f) Authenticity | - authentic (S1; E9)<br>- inauthentic, misleading (S2; E3) |
| (g) Appearance | - attractive (S4; E2)<br>- think they are attractive (S2) |
| **4. Miscellaneous views** | - too much time available (S1)<br>- take credit instead of photographer (E1)<br>- image of self met their standards (E1) |

or validation. A substantial number also cited perceptions of narcissism ($n = 7$) or specific related traits ($n = 44$), such as being self-absorbed or conceited. In relation to elsies, references to attention-seeking or validation motives ($n = 9$) or narcissism-related traits ($n = 4$) were considerably less frequent. Additionally, beliefs about society's perceived motives for elsie-posters were more diverse, citing motives such as curating self-presentation, documenting experiences, or simply sharing aspects of oneself.

On average, participants rated society's attitudes about selfies to be significantly less positive ($M = 4.0$, $SD = 1.2$) than elsies ($M = 3.0$, $SD = 1.2$), $t(124) = 8.48$, $p < .001$. The mean rating for selfie-posters fell near the midpoint of the scale, suggesting a relatively neutral perception rather than a strongly positive or negative one. Participants expressed a range of perceived societal attitudes toward both selfies and elsies, suggesting that self-presentation through either format was

associated with both positive and negative attitudes. Some participants offered general impressions without elaborating on their reasoning; of these, elsies were more frequently described as "normal" ($n = 11$) or socially acceptable ($n = 4$) compared to selfies ($n = 3$). A few noted that societal perceptions of selfies ($n = 4$) and elsies ($n = 4$) are context-dependent and vary by the individual posting. Among those who cited specific characteristics of posters perceived by society, responses about elsie-posters tended to reflect either clearly positive or negative perceptions, whereas 11 participants described mixed views of selfie-posters. Although perceptions revealed both negative and positive connotations, positive perceptions for both image types were relatively consistent in content: selfie-posters were most seen as confident ($n = 31$), while elsie-posters were often associated with having a strong social life ($n = 31$). Interestingly, a distinct theme emerged for elsies, with a few participants ($n = 4$) suggesting that societal perceptions may also extend to the photographer or to contextual aspects of the image itself.

## Causal attributions for social feedback

The comparison of mean valence of feedback revealed a significant main effect of feedback condition, $F(1, 119) = 363.32$, $p < .001$, $\eta_p^2 = .75$. No other effects were significant. On average, the favorable feedback condition ($M = 5.8$, $SD = 0.9$) reported receiving more positive feedback compared to the unfavorable feedback condition ($M = 2.4$, $SD = 1.0$), indicating that the manipulation was successful.

The comparison of mean locus of control revealed a significant main effect of image condition, $F(2, 118) = 6.71$, $p = .002$, $\eta_p^2 = .10$. According to Tukey's HSD, those who posted a selfie ($M = 4.3$, $SD = 1.3$) or an elsie ($M = 4.1$, $SD = 1.7$) were both more likely to attribute the feedback received to internal factors compared to those who posted a meme ($M = 3.1$, $SD = 1.9$), $p = .002$ and $p = .022$, respectively. The themes observed for the causal attributions, presented in Table 2, were consistent with the quantitative findings. As shown in Table 3, all attributions for memes were linked to external factors, with feedback largely explained in terms of audience reception. Unfavorable meme feedback was often attributed to the audience being offended, holding different views, or finding the content unrelatable, while favorable feedback was linked to the meme perceived as relatable or humorous. In contrast, selfies and elsies prompted predominately internal attributions, such as personal appearance, expression, or self-presentation choices. Specifically, 71% of attributions provided for selfies and elsies were related to internal factors, with many related to their appearance. Notably, individuals were more likely to attribute favorable feedback to controllable internal factors for both selfies (controllable: $n = 9$; uncontrollable: $n = 5$) and elsies (controllable: $n = 8$; uncontrollable: $n = 4$). Unfavorable attributions, in contrast, were more frequently linked to uncontrollable internal factors for selfies (controllable: $n = 6$; uncontrollable: $n = 8$) and elsies (controllable: $n = 4$; uncontrollable: $n = 9$). Overall, controllable internal factors cited hair, clothing, accessories, facial expression, and body language or pose. Uncontrollable internal factors involved aspects such as general attractiveness or appearance, skin complexion, weight, authenticity, being female, humor or playfulness, tiredness, or general personal qualities. Despite their prevailing negative society view of selfies, participants did not attribute feedback to societal perceptions for this type of self-presentation.

No participants cited internal factors for feedback for memes. The values in parentheses indicate the number of participants who referenced the content category for the attribution. Attributions without accompanying values were referenced by a single participant. One participant within the selfie and elsie conditions provided two internal attributions to explain feedback.

## Discussion

The present study examines the attributions individuals made for favorable and unfavorable feedback received in response to selfies, elsies, or memes posted on Instagram. The broad themes and nuances revealed contribute to a better understanding of how individuals negotiate social perception and self-presentation on SNSs. Selecting selfies or elsies to share on Instagram can render the self more salient and lead individuals to attribute outcomes, favorable or not, more

**Table 2.** Internal causal attributions cited for favorable and unfavorable feedback for selfies and elsies.

| Overarching Theme Subthemes | Selfies | | Elsies | |
|---|---|---|---|---|
| | Favorable (n = 20) | Unfavorable (n = 23) | Favorable (n = 20) | Unfavorable (n = 20) |
| **Theme 1: Physical appearance elements (uncontrollable)** | | | | |
| General appearance evaluation | - appearance-related adequacy | - appearance-related<br>- appearance-related inadequacy (2) | - appearance-related adequacy | - appearance-related inadequacy (2)<br>- unflattering<br>- unappealing |
| Focus on specific feature | - female | - poor skin complexion<br>- face<br>- lashes | | - weight-related (2)<br>- skin complexion (2)<br>- skin color |
| Perceived presentation adequacy | | - appeared tired<br>- unpolished appearance (2) | - clothing | - unprepared<br>- appeared tired |
| **Theme 2: Outward behavior (controllable)** | | | | |
| Expression | - smiling (3)<br>- happy | - no smile (3)<br>- general expression<br>- awkward | - smile (5)<br>- funny expression | - unhappy |
| Gestures/ Posture | - pose | | - pose | - pose<br>- body language |
| Interpersonal style | - hair (2)<br>- clothing<br>- accessories | - clothing | | - accessories |
| **Theme 3: Personal Qualities (uncontrollable)** | | | | |
| Impression conveyed | - authentic<br>- humor | | - fun<br>- unspecified personal quality<br>- authentic | |

readily to internal over external factors [25]. In contrast, memes are a form of self-expressive but not self-focused content. Since the humour is the primary focus, individuals can readily attribute unfavorable feedback to flaws in the content itself and audience reception (i.e., uncontrollable external factors) rather than to themselves.

According to the self-serving attribution bias, people interpret the behaviour of others in ways that are favorable to the self, even though such interpretations may not necessarily be accurate [30]. Thus, to protect one's self-esteem, people are postulated to attribute positive events to their own character but attribute negative events to external factors [31]. For example, they may attribute a favourable outcome to their ability but view themselves as unlucky when the outcome is unfavorable [30]. This self-protection mechanism has been demonstrated across performance-related domains, such as task success [32,33], athletic performance [34], work performance [35], and peer-comparison tasks [36]. Similar patterns were observed in Stsiampkouskaya et al.'s study [5], in which participants attributed others' imagined reactions to social media feedback in a way that maintained a favorable view of themselves. In the current study, however, contrary to pre-dictions based on the self-serving bias, participants' attributions for what could be perceived as failures associated with their selfies and elsies were predominantly internal. These findings indicate that attributional patterns may shift when the feedback domain is appearance-based, especially for women in Western cultural settings, who represented three-quarters of the sample.

Women in cultures that place a strong emphasis on appearance are particularly likely to become objectified – treated as objects to be looked at and evaluated based on how they look [37,38]. Over time, they internalize this external per-spective and start viewing and judging their physical appearance from a third-person perspective. Research shows that contexts prompting individuals to consider how others will evaluate their appearance reliably heighten self-objectification

**Table 3. External causal attributions cited for favorable and unfavorable feedback for selfies/elsies compared to memes.** Note. For selfies and elsies, the coding in parentheses identifies if the content category for the attribution was cited in reference to selfies (S) or elsies (E). The values in parentheses indicate the number of participants who referenced the attribution. 11 participants did not provide attributions for the feedback (5 selfies, 2 elsies, and 1 meme). Four participants provided two external attributions for the feedback (2 selfies, 1 elsies, and 1 meme).

| Overarching Theme Subtheme | Selfies/ Elsies | | Memes | |
|---|---|---|---|---|
| | Favorable (n = 40) | Unfavorable (n = 43) | Favorable (n = 22) | Unfavorable (n = 20) |
| **Theme 1: Audience Characteristics** | | | | |
| Relation to imagined audience | - friends (S2)<br>- peers (S2)<br>- personal relationships (E2) | - strangers (S1)<br>- peers (S1) | | |
| Qualities | - supportive (E1)<br>- nice (E1)<br>- polite (E2) | - low self-confidence (S1; E1) | - acknowledge their posts (1) | |
| **Theme 2: Context** | | | | |
| Context | - quality of image (S1)<br>- timing (S1)<br>- hashtags used (E1) | - location (S1)<br>- time restraint (S1)<br>- no context (E1) | | |
| **Theme 3: Content** | | | | |
| Audience reception | - well-liked (E1) | - personal opinion (E1) | - relatable (10)<br>- same opinion (1) | - not relatable (3)<br>- different opinion (6)<br>- different sense of humor (1)<br>- perceived as egotistical (1)<br>- offended (4) |
| Quality of content | | - irrelevant (S1; E1) | - funny (7)<br>- positive (1)<br>- outdated (1) | - focus of joke (2)<br>- foul language (1) |
| Visual presentation quality | - image quality (S2) | - image quality (S1; E2) | - image quality (1) | |

during SNS use [39]. Experimental research reveals that the act of taking selfies, regardless of whether they are shared, can increase state self-objectification and body dissatisfaction compared to taking photos of objects [26], among men and women [40]. Self-objectification has also been shown to increase after receiving favorable appearance-based comments on their selfies [41]. Given that (women) participants believed their selfie or elsie would be shared with an audience comprised of their peers and evaluated for feedback, the study context likely activated this objectifying gaze. As a result, internal causal attributions for social feedback were largely centered on their physical appearance.

Moreover, participants more frequently attributed favorable feedback to controllable factors, whereas unfavorable feedback was often linked to factors beyond their control. Individuals appeared to struggle with internalizing praise, seeing positive feedback as conditional or situational (e.g., hair, smiling expression, accessories, and clothing) rather than as a reflection of enduring personal qualities. Unfavorable responses appear to resonate more deeply, being interpreted as confirmation of perceived personal shortcomings (e.g., unappealing appearance, skin complexion, and weight). Such attributional tendencies align closely with patterns associated with shame. When negative appearance feedback is received, it is more likely to evoke shame, an emotion rooted in the perception that something is wrong with the whole self rather than with a specific behavior [42]. Shame reliably promotes an attributional pattern characterized by global and uncontrollable causes – perceiving negative outcomes as stemming from broad, stable, and unchangeable personal flaws (e.g., "I'm unattractive," "There is something wrong with me") rather than specific, situational factors (e.g., poor lighting or camera angle). These global and uncontrollable attributions have been documented in body and appearance contexts specifically [e.g., 43].

Notably, although participants commonly believed that society holds relatively negative views toward those who post selfies, none attributed the unfavorable feedback they received to this broader social norm. A common theme was that selfie-posters were perceived by society as exhibiting narcissistic tendencies, where as elsie-posters are viewed as highly sociable. Given that distinct perceptions of self-presentational content on SNSs have persisted for nearly a decade, repeated exposure to prevailing cultural narratives likely contribute to individuals' schema about how society views those who post selfies versus elsies. Even individuals who personally reject such generalizations, such as those who noted that perceptions depend on the person posting, may still assume that most people endorse them, given the frequency with which these narratives are reinforced in media discourse. Objectification theory suggests that the internalized perspective manifests as habitual and vigilant self-monitoring of their appearance for flaws [37,38]. Thus, even when aware of negative societal attitudes toward selfie-posting, individuals may default to appearance-focused internal attributions when their image is evaluated. Additionally, research on the imagined audience shows that social media users think primarily in terms of specific viewers rather than society at large, which is perceived as too distant to be causally relevant [44]. Participants may have viewed the immediate audience, not cultural stereotypes, as the primary evaluators, reducing the likelihood that societal narratives were used as self-protective explanations.

Overall, these findings highlight how selfies and elsies shared on SNSs not only make appearance salient but also fosters internalized appearance evaluation through the nature of the feedback received. When the self is viewed as an object under gaze, situational defences might be less salient or psychological accessible, and traditional self-serving logic may not hold, increasing the likelihood of internalized negative attributions.

Why might individuals be quick to infer that others hold a negative view of their appearance? According to self-verification theory, people prefer others to perceive them as they see themselves. Thus, they process feedback about themselves in ways that align with their self-views, even if their self-views are negative [45]. Young women are particularly susceptible to internalizing the thin ideal – a sociocultural standard that equates thinness with attractiveness and success – due to frequent exposure to idealized media portrayals and heightened self-evaluative concerns during identity development [46]. Given that the sample consisted primarily of young women, it is plausible that many participants held self-perceptions that fell short of societal standards of attractiveness. Consequently, they may have projected these views onto their imagined audience. Alternatively, attributions for unfavorable feedback to internal controllable factors may arise from the positivity bias on Instagram, wherein favorable reactions are the norm. A sample of U.S.-American adolescents revealed that 91% rarely, if ever, received negative feedback on their posts [47]. Within this social context, unfavorable feedback may stand out as especially diagnostic, prompting individuals to explain it by attributing it to fixed, less desirable traits they perceive in themselves.

Across both favorable and unfavorable outcomes, participants overwhelmingly attributed responses to meme posts to external factors. This pattern for meme-related feedback can be understood through the lens of online disinhibition. According to the online disinhibition effect, anonymity and invisibility fosters psychological distance and a sense that one's online actions are detached from the real self [48]. In this state, users experience dissociation between one's online behaviour and one's offline identity, perceived as what goes on online is less reflective of the real self, resulting in less emotional investment and more detached from outcomes. For the meme condition, participants had minimal involvement in creating the content, contributing only by selecting a pre-existing meme, and the posts were shared without identifying information. The meme became a non-self-referential object, which according to the literature would generate low self-objectification [26,40], resulting in lower personal accountability for reactions. As a result, participants likely experienced the meme as an object independent of the self, making audience reactions feel directed at the humor of the meme rather than at them personally. This sense of detachment was likely amplified by anonymity, which can reduce the identity stake in outcomes [48]. Consequently, participants may have perceived the imagined audience, not themselves, as the central agent in evaluating the post, prompting participants to externalize the source of the feedback they received.

## Limitations and future directions

This study was exploratory in nature, aiming to document homogeneity and heterogeneity of causal attributions individuals make in response to social feedback on different types of posts, without directly testing any specific theory. The findings suggest that self-images can trigger processes such as self-objectification and self-verification, and memes trigger online disinhibition processes. These may shape the type of attributions made for social feedback. While the study does not offer definitive tests of these mechanisms, it provides new and compelling leads for future research to investigate using more targeted, theory-driven methodologies.

To simulate a SNS experience, the study asked participants to contribute to a post (selfie, elsie, or meme), framed as being shared on Instagram with a broad and relevant audience of peers. However, the identities of those giving feedback were not visible, leaving participants to imagine the audience and interpret feedback in the absence of specific social cues. In reality, when posting on their Instagram feed, individuals have a clearer sense of their target audience and expectations about how certain types of self-presentation are received. This familiarity may allow for more tailored attributions based on who responds and how. Thus, while the current findings offer valuable insight into attribution processes in ambiguous social situations, future research should explore how these processes unfold in more ecologically valid settings.

As previously noted, the results must be considered in the context of the sample characteristics, particularly the over-representation of women who comprised 76% of the participants. Prior research suggests that women, particularly in Western cultural contexts, experience higher levels of self-objectification, engage more frequently in appearance-based self-evaluation, and may respond differently to both positive and negative appearance feedback than men. As a result, the attributional patterns observed in this study may reflect gendered experiences of social media use and appearance-related evaluation. Future research should examine whether similar patterns emerge in more gender-balanced samples or in samples that allow for meaningful comparisons across gender and/or cultural identities.

## Conclusion

This study found that selecting selfies and elsies to share on Instagram led individuals to attribute both favorable and unfavorable outcomes more readily to internal rather than external factors. When the physical self is at the center of self-presentation, individuals may project their own insecurities and self-perceptions onto the imagined audience, assuming that others also view their appearance as falling short of idealized societal standards. Even when feedback was favorable, it was often attributed to surface-level or easily changeable aspects—such as facial expression, clothing, or accessories—rather than to enduring qualities of physical appearance. In contrast, memes – non-self-referent posts – invited greater psychological distancing, more external attributions, and less personal investment. Together, the results contribute to a deeper understanding of how self-presentation and audience perception interact to influence users' interpretations of social feedback on SNS platforms.

The implications of self-objectification, self-verification, and psychological connection to the posts are noteworthy. When individuals perceive unfavorable feedback on self-focused posts as confirmation of negative self-beliefs of their attractiveness, this may reinforce insecurities and contribute to diminished self-esteem or well-being over time. Understanding these dynamics is crucial in promoting healthier social media use, as it suggests that efforts to buffer users from negative affect may require more than encouraging positive posting strategies—they must also address the underlying self-concept and how individuals interpret social feedback in relation to it.

## Author contributions

**Conceptualization:** Malinda Desjarlais.

**Data curation:** Malinda Desjarlais.

**Formal analysis:** Malinda Desjarlais.

**Investigation:** Malinda Desjarlais.

**Methodology:** Malinda Desjarlais.

**Project administration:** Malinda Desjarlais.

**Supervision:** Malinda Desjarlais.

**Writing – original draft:** Malinda Desjarlais.

**Writing – review & editing:** Malinda Desjarlais.

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
