## [Decision Letter · Decision Letter 0]

22 Oct 2025

PONE-D-25-26407It's not you, it's me: A thematic analysis of written causal attributions for favorable and unfavorable feedback on different types of Instagram image sharingPLOS ONE

Dear Dr. Malinda.

Thank you for submitting your manuscript to PLOS ONE. After careful consideration, we feel that it has merit but does not fully meet PLOS ONE’s publication criteria as it currently stands. Therefore, we invite you to submit a revised version of the manuscript that addresses the points raised during the review process.

Kindly go through the suggested revisions

Please submit your revised manuscript by Dec 06 2025 11:59PM. If you will need more time than this to complete your revisions, please reply to this message or contact the journal office at plosone@plos.org. Please include the following items when submitting your revised manuscript:

We look forward to receiving your revised manuscript.

Kind regards,

Shiva Ram Male, M.S, PG.Dipl, PhD

Academic Editor

PLOS ONE

Journal Requirements:

3. We note that there is identifying data in the Supporting Information file <file name>. Due to the inclusion of these potentially identifying data, we have removed this file from your file inventory. Prior to sharing human research participant data, authors should consult with an ethics committee to ensure data are shared in accordance with participant consent and all applicable local laws.

-Location data

Additional Editor Comments (if provided):

Dear Author,

Kindly go through the revisions suggested by the Reviewer

Reviewers' comments:

Reviewer's Responses to Questions

**Comments to the Author**

1. Is the manuscript technically sound, and do the data support the conclusions?

Reviewer #1: Yes

2. Has the statistical analysis been performed appropriately and rigorously? 

Reviewer #1: Yes

3. Have the authors made all data underlying the findings in their manuscript fully available?

Reviewer #1: Yes

4. Is the manuscript presented in an intelligible fashion and written in standard English?

Reviewer #1: Yes

5. Review Comments to the Author

Reviewer #1: your manuscript titled “It’s not you, it’s me: A thematic analysis of written causal attributions for favorable and unfavorable feedback on different types of Instagram image sharing.” This is a timely and thoughtfully designed study that offers valuable insights into how individuals interpret social feedback on self-presentational content across social media platforms. Below are my comments and suggestions aimed at helping you refine and strengthen your manuscript.

Major Strengths:

Novel Contribution: The manuscript extends attribution theory into the domain of social media feedback, which is relatively underexplored. The focus on real-time personal reactions rather than imagined scenarios is a significant advancement.

Methodological Rigor: The experimental use of fabricated feedback and the inclusion of three image types (selfie, elsie, meme) is innovative and well-executed.

Thematic Clarity: The reflexive thematic analysis is robust, with well-categorized themes and illustrative participant responses.

Integration with Theory: The use of attribution theory, sociometer theory, self-objectification, and self-verification theories helps provide depth to your findings.

Suggestions for Improvement:

1. Theoretical Clarity: While self-serving bias is mentioned, the observed deviation (internal attributions even for unfavorable feedback) could benefit from deeper theoretical reflection. Could self-objectification override typical self-protective mechanisms? A more direct engagement with the contradiction between attribution theory and observed patterns would strengthen your discussion.

2. Sample and Generalizability: sample (majority young females) is clearly reported. However, consider expanding the discussion on how this demographic bias may have influenced the attribution patterns, particularly in relation to appearance-based evaluations and societal beauty norms.

3. Interpretation of Meme Feedback: The findings around meme feedback (predominantly external attributions) are intriguing. You may consider further elaboration using the online disinhibition effect or content detachment frameworks to explain this distancing.

4. Perceived Societal Attitudes vs. Attribution: A strong point of your study is that participants acknowledged negative societal perceptions toward selfies but did not attribute unfavorable feedback to those broader norms. This disconnect is important and could be more explicitly explored in the discussion section.

Minor Recommendations:

1. Standardize terminology: consider consistently using either “self-images” or “selfies and elsies” for clarity.

2. The methods section includes some repetition (e.g., feedback condition explanations); tightening this would improve flow.

3. Please proofread for minor typographical issues (e.g., "appearancerelated" should be "appearance-related").

4. A procedural diagram illustrating the study design could enhance reader comprehension.

5. If possible, consider sharing your thematic codebook in the Supporting Information for transparency and reproducibility.

Final Remarks i felt:

Overall, this is a well-written and engaging manuscript that makes a meaningful contribution to the literature on digital self-presentation and social cognition. I believe it will be of great interest to both psychological and media studies audiences. I recommend very minor revision to address the points above, which will further enhance the clarity and impact of your work.

Best regards,

Reviewer

6. PLOS authors have the option to publish the peer review history of their article (what does this mean?). If published, this will include your full peer review and any attached files.

Reviewer #1: **Yes:** Dr. P. Phani Krishna

---

## [Author Response · Author response to Decision Letter 1]

1 Dec 2025

A copy of the comments and responses has been uploaded to the system. The following is a copy of that document:

Section 1 - Editor Comments

Journal Requirements:

Response: The manuscript and file naming has been checked to ensure it meets PLOS ONES’s style requirements.

Response: the following (in red) has been added to the manuscript (pages 10-11) to clarify the consent process:

Participants arrived at the lab. Participants completed an informed consent process and subsequently provided written consent. In accordance with the institution’s ethical guidelines, university students, irrespective of age, may independently provide informed consent for research participation. Accordingly, parental consent was not required for participants under 18 years of age.

3. We note that there is identifying data in the Supporting Information file <file name>. Due to the inclusion of these potentially identifying data, we have removed this file from your file inventory. Prior to sharing human research participant data, authors should consult with an ethics committee to ensure data are shared in accordance with participant consent and all applicable local laws.

-Location data

Response: After reviewing the content of the supplementary information, I felt that it did not contribute meaningfully to the manuscript. These documents simply summarized the characteristics of the participants and their quantitative responses to the questions in a different format that what was presented in the manuscript. Therefore, I decided to remove all the supplementary information documents. If researchers require access to the data to verify the findings the data would be made available upon request (as outlined below).

Response: A complete data file containing all participant responses cannot be shared publicly for ethical reasons. The consent form that participants signed stated that “Your individual responses will not be published in any form.” To comply with the standards approved by the Mount Royal University Human Research Ethics Board, the data file can only be provided upon request by emailing the author directly. These requests must go through the author because Mount Royal University does not have an office responsible for managing such data access.

Response: As Supporting Information has been removed, captions are no longer needed.

Response: not applicable

Response: The references were reviewed and ensure correct.

Additional Editor Comments (if provided):

Dear Author,

Kindly go through the revisions suggested by the Reviewer

Section 2 - Reviewers' comments:

Reviewer's Responses to Questions

Comments to the Author

5. Review Comments to the Author

Reviewer #1: your manuscript titled “It’s not you, it’s me: A thematic analysis of written causal attributions for favorable and unfavorable feedback on different types of Instagram image sharing.” This is a timely and thoughtfully designed study that offers valuable insights into how individuals interpret social feedback on self-presentational content across social media platforms. Below are my comments and suggestions aimed at helping you refine and strengthen your manuscript.

Major Strengths:

Novel Contribution: The manuscript extends attribution theory into the domain of social media feedback, which is relatively underexplored. The focus on real-time personal reactions rather than imagined scenarios is a significant advancement.

Methodological Rigor: The experimental use of fabricated feedback and the inclusion of three image types (selfie, elsie, meme) is innovative and well-executed.

Thematic Clarity: The reflexive thematic analysis is robust, with well-categorized themes and illustrative participant responses.

Integration with Theory: The use of attribution theory, sociometer theory, self-objectification, and self-verification theories helps provide depth to your findings.

Response: Thank you for your support.

Suggestions for Improvement:

1. Theoretical Clarity: While self-serving bias is mentioned, the observed deviation (internal attributions even for unfavorable feedback) could benefit from deeper theoretical reflection. Could self-objectification override typical self-protective mechanisms? A more direct engagement with the contradiction between attribution theory and observed patterns would strengthen your discussion.

Response: Thank you for pointing this out—I agree that the discussion would benefit from a deeper theoretical reflection. In response, I have expanded the discussion to more thoroughly address the theoretical implications of the findings. The following additions were incorporated into the discussion to strengthen this section:

First, the discussion of the pattern “attributing favorable feedback to controllable factors and unfavorable to uncontrollable” was expanded. The following was added to the discussion (pages 20-21):

Such attributional tendencies align closely with patterns associated with shame. When negative appearance feedback is received, it is more likely to evoke shame, an emotion rooted in the perception that something is wrong with the whole self rather than with a specific behavior [43]. Shame reliably promotes an attributional pattern characterized by global and uncontrollable causes – perceiving negative outcomes as stemming from broad, stable, and unchangeable personal flaws (e.g., “I’m unattractive,” “There is something wrong with me”) rather than specific, situational factors (e.g., poor lighting or camera angle). These global and uncontrollable attributions have been documented in body and appearance contexts specifically [e.g., 44].

To build on this point, the discussion of participants’ non-use of societal attitudes toward selfies was moved earlier in the discussion section. This shift allowed for a more coherent and theoretically grounded explanation of how self-objectification may have overshadowed typical self-protective mechanisms in this context. Accordingly, the following material was added to the manuscript immediately after the non-use of societal attitudes section of the discussion (pages 21-22):

Objectification theory suggests that the internalized perspective manifests as habitual and vigilant self-monitoring of their appearance for flaws [37, 38]. Thus, even when aware of negative societal attitudes toward selfie-posting, individuals may default to appearance-focused internal attributions when their image is evaluated. Additionally, research on the imagined audience shows that social media users think primarily in terms of specific viewers rather than society at large, which is perceived as too distant to be causally relevant [45]. Participants may have viewed the immediate audience, not cultural stereotypes, as the primary evaluators, reducing the likelihood that societal narratives were used as self-protective explanations.

Overall, these findings highlight how selfies and elsies shared on SNSs not only makes appearance salient but also fosters internalized appearance evaluation through the nature of the feedback received. When the self is viewed as an object under gaze, situational defences might be less salient or psychological accessible, and traditional self-serving logic may not hold, increasing the likelihood of internalized negative attributions.

2. Sample and Generalizability: sample (majority young females) is clearly reported. However, consider expanding the discussion on how this demographic bias may have influenced the attribution patterns, particularly in relation to appearance-based evaluations and societal beauty norms.

Response: Although the sample composition (76% women) was briefly acknowledged in the original discussion, I agree that this demographic imbalance could have influenced participants’ causal attributions, particularly given the relevance of self-objectification theory to appearance-based evaluations. In response, I have integrated this point more explicitly earlier in the discussion. The paragraph (page 20) has been revised accordingly, with the new additions highlighted in red:

Women in cultures that place a strong emphasis on appearance are particularly likely to become objectified – treated as objects to be looked at and evaluated based on how they look [37, 38]. Over time, they internalize this external perspective and start viewing and judging their physical appearance from a third-person perspective. Research shows that contexts prompting individuals to consider how others will evaluate their appearance reliably heighten self-objectification during SNS use [39-41]. Experimental research reveals that the act of taking selfies, regardless of whether they are shared, can increase state self-objectification and body dissatisfaction compared to taking photos of objects [26], among men and women [41]. Self-objectification has also been shown to increase among women after receiving favorable appearance-based comments on their selfies [42]. Given that (women) participants believed their selfie or elsie would be shared with an audience comprised of their peers and evaluated for feedback, the study context likely activated this objectifying gaze. As a result, internal causal attributions for social feedback were largely centered on their physical appearance.

Additionally, the sample composition is now acknowledged as a limitation of the study. The following was added to the Limitations and future directions section (page 24):

As previously noted, the results must be considered in the context of the sample characteristics, particularly the overrepresentation of whom, who comprised 76% of the participants. Prior research suggests that women, particularly in Western cultural contexts, experience higher levels of self-objectification, engage more frequently in appearance-based self-evaluation, and may respond differently to both positive and negative appearance feedback than men. As a result, the attributional patterns observed in this study may reflect gendered experiences of social media use and appearance-related evaluation. Future research should examine whether similar patterns emerge in more gender-balanced samples or in samples that allow for meaningful comparisons across gender and/or cultural identities.

3. Interpretation of Meme Feedback: The findings around meme feedback (predominantly external attributions) are intriguing. You may consider further elaboration using the online disinhibition effect or content detachment frameworks to explain this distancing.

Response: Although the online disinhibition effect was mentioned in the original discussion, I agree that it required deeper theoretical consideration in the context of the causal attributions for the meme condition. In response, the section dedicated to the findings for the meme condition has been revised to more fully explain the underlying mechanisms of the online disinhibition effect and how these mechanisms may account f

---

## [Editor Report · Decision Letter 1]

20 Mar 2026

It's not you, it's me: A thematic analysis of written causal attributions for favorable and unfavorable feedback on different types of Instagram image sharing

PONE-D-25-26407R1

Dear Dr.Malinda,

We’re pleased to inform you that your manuscript has been judged scientifically suitable for publication and will be formally accepted for publication once it meets all outstanding technical requirements.

Kind regards,

Shiva Ram Male, M.S, PG.Dipl, PhD

Academic Editor

PLOS One
---

## [Editor Report · Acceptance letter]

PONE-D-25-26407R1

PLOS One

Dear Dr. Desjarlais,

I'm pleased to inform you that your manuscript has been deemed suitable for publication in PLOS One. Congratulations! Your manuscript is now being handed over to our production team.

Kind regards,

on behalf of

Dr. Shiva Ram Male

Academic Editor

PLOS One